# Sex-specific attenuation of photoreceptor degeneration by reserpine in a rhodopsin P23H rat model of autosomal dominant retinitis pigmentosa

Hyun Beom Song[1,2†], Laura Campello[1†‡], Anupam Mondal[1†], Holly Y Chen[1§], Milton A English[1], Michael Glen[1], Phillip Vanlandingham[3], Rafal Farjo[3], Anand Swaroop[1]*

[1]Neurobiology, Neurodegeneration and Repair Laboratory, National Eye Institute, National Institutes of Health, Bethesda, United States; [2]Department of Biomedical Sciences, Seoul National University College of Medicine, Seoul, Republic of Korea; [3]EyeCRO, Oklahoma City, United States

*For correspondence:
swaroopa@nei.nih.gov

†These authors contributed equally to this work

Present address: ‡Avista Therapeutics, Pittsburgh, United States; §Department of Cell, Developmental and Integrative Biology, University of Alabama at Birmingham, Birmingham, United States

## eLife Assessment

This **important** Research Advance presents **compelling** evidence on the neuroprotective effects of reserpine in a well-established model of retinitis pigmentosa (P23H-1). This study builds on previous work establishing reserpine as a neuroprotectant in models of Leber congenital amaurosis. Here authors show reserpine's disease gene-independent influence on photoreceptor survival and emphasizes the importance of considering biological sex in understanding inherited retinal degeneration and the impact of drug treatments on mutant retinas. The work will be of interest to vision researchers as well as a broad audience in translational research.

**Abstract** Inherited retinal degenerations (IRDs) constitute a group of clinically and genetically diverse vision-impairing disorders. Retinitis pigmentosa (RP), the most common form of IRD, is characterized by gradual dysfunction and degeneration of rod photoreceptors, followed by the loss of cone photoreceptors. Recently, we identified reserpine as a lead molecule for maintaining rod survival in mouse and human retinal organoids as well as in the *rd16* mouse, which phenocopy Leber congenital amaurosis caused by mutations in the cilia-centrosomal gene *CEP290* (Chen et al., 2023). Here, we show the therapeutic potential of reserpine in a rhodopsin P23H rat model of autosomal dominant RP. At postnatal day (P) 68, when males and females are analyzed together, the reserpine-treated rats exhibit higher rod-derived scotopic b-wave amplitudes compared to the controls with little or no change in scotopic a-wave or cone-derived photopic b-wave. Interestingly, the reserpine-treated female rats display enhanced scotopic a- and b-waves and photopic b-wave responses at P68, along with a better contrast threshold and increased outer nuclear layer thickness. The female rats demonstrate better preservation of both rod and cone photoreceptors following reserpine treatment. Retinal transcriptome analysis reveals sex-specific responses to reserpine, with significant upregulation of phototransduction genes and proteostasis-related pathways, and notably, genes associated with stress response. This study builds upon our previously reported results reaffirming the potential of reserpine for gene-agnostic treatment of IRDs and emphasizes the importance of biological sex in retinal disease research and therapy development.

## Introduction

Inherited retinal degenerations (IRDs) are a diverse group of vision-deteriorating disorders caused by mutations in over 280 genes (*Duncan et al., 2018*; RetNet, https://retnet.org; August 2024). Despite their genetic diversity, these disorders share a common outcome, which is the gradual loss of photoreceptor cells (*Farrar et al., 2017*; *Swaroop et al., 2010*; *Wright et al., 2010*). Among the more severe forms of IRDs is Leber congenital amaurosis (LCA), which is characterized by vision loss congenitally or during early infancy. The retinal ciliopathy caused by mutations in *CEP290* accounts for 20–25% of LCA (*Coppieters et al., 2010*; *Leroy et al., 2021*). On the other hand, retinitis pigmentosa (RP) is the most common form of IRDs, affecting 1 in 3000–4000 individuals, and characterized by the primary degeneration of rod photoreceptors, followed by the loss of cone photoreceptors (*Cross et al., 2022*; *Farrar et al., 2017*; *Verbakel et al., 2018*). The initial symptom is reduced night vision, which is followed by a gradual constriction of the visual field, often resulting in tunnel vision. Mutations in the rhodopsin (*RHO*) gene, encoding the light-sensing visual pigment essential for phototransduction in rod photoreceptors, are a major cause of retinal degeneration (*Athanasiou et al., 2018*), with the P23H change accounting for over 10% of autosomal dominant RP (adRP) cases in the US (*Sullivan et al., 2006*). Notably, visual function at the macula, the cone-dominant central region, is relatively preserved until later stages of the disease in many IRDs (*Cideciyan and Jacobson, 2019*; *Cross et al., 2022*). The precise mechanisms through which mutations cause photoreceptor cell death in IRDs are not yet fully understood. Nevertheless, studies using animal models have shed light on several pathways involved in photoreceptor cell death. These include abnormal photoreceptor outer segment morphogenesis, mitochondrial dysfunction, unfolded protein responses associated with the accumulation of misfolded protein in the endoplasmic reticulum, metabolic overload, aberrant protein trafficking, and chronic activation of phototransduction, among others (*Bales and Gross, 2016*; *Chen et al., 2022*; *Jiang et al., 2022*; *Lobanova et al., 2013*; *Pierce, 2001*; *Power et al., 2020*; *Won et al., 2009*).

Therapeutic options are currently limited for IRDs. To date, the only approved treatment is a gene replacement therapy for one form of LCA caused by mutations in the gene *RPE65* (*Russell et al., 2017*). Extensive genetic heterogeneity of IRDs poses a significant challenge for developing gene-specific therapies, which in turn restricts the broad applicability of such interventions across a range of patient populations. Most IRD mutations are rare and often limited to a small number of patients or families; therefore, developing targeted interventions is less feasible due to the associated costs. In contrast, gene-agnostic therapies can reach larger patient cohorts. LaVail, Steinberg and colleagues were among the first to show therapeutic potential of many different neuroprotective factors in animal models of IRDs (*LaVail et al., 1992*). Since then, many different neurotrophic factors, such as ciliary neurotrophic factor, rod-derived cone viability factor and pigment epithelium-derived factor, have been used in animal models and/or clinical trials with promising results (*Byrne et al., 2015*; *Do Rhee et al., 2022*; *Michelis et al., 2021*; *Wen et al., 2012*). Neuroprotection including the use of antioxidant and anti-apoptotic compounds is predicted to slow down the rate of degeneration and preserve retinal function (*Leinonen et al., 2024*; *Pinilla et al., 2022*; *Tolone et al., 2023*; *Yao et al., 2022*). Many other gene-agnostic therapeutic paradigms, including retinal stimulation and optogenetic strategies, are under investigation for IRDs (*John et al., 2022*; *Roska, 2019*; *Sahel et al., 2021*; *Yue et al., 2016*). However, numerous hurdles, including delivery methods and stimulation techniques, remain.

Given that photoreceptor cell death is the final outcome and the cause of vision loss in all IRDs (*Wright et al., 2010*), we had proposed identification of therapeutic targets by examining the convergence of molecular pathways induced by divergent disease-causing IRD mutations (*Swaroop et al., 2010*; *Yu et al., 2004*). Molecular analyses of animal models have provided new insights into distinct cellular pathways that can be modulated for treating IRDs (*Jiang et al., 2022*; *Tolone et al., 2023*). We recently reported an unbiased large screen of small molecules based on survival of rod photoreceptors using retinal organoids derived from induced pluripotent stem cells of *rd16* mice (*Chen et al., 2023*), which phenocopies *CEP290*-LCA (*Chang et al., 2006*). Reserpine, the lead molecule identified by this screen, augmented rod survival in *CEP290*-LCA patient-derived retinal organoids and as well as in the *rd16* mouse retina in vivo (*Chen et al., 2023*). The neuroprotective effect of reserpine was achieved, at least in part, through the restoration of the balance between autophagy and the ubiquitin-proteasome system, the stress-response pathways maintaining cellular proteostasis (*Chen et al., 2023*; *Villarejo-Zori et al., 2021*; *Weinberg et al., 2022*). In this study, we evaluated

the therapeutic potential of reserpine for adRP caused by the *RHO* P23H mutation, which reportedly causes protein misfolding and secondary proteasome defect leading to photoreceptor cell death (*Illing et al., 2002*; *Kaushal and Khorana, 1994*; *Qiu et al., 2019*; *Sakami et al., 2011*). Using the P23H-1 rat model, we performed a series of assays to investigate the response to reserpine treatment on retinal pathology. We demonstrate that reserpine preserved photoreceptor cell survival and retinal function, with a more pronounced effect observed in female rats. Consistent with our previous study showing the efficacy of reserpine for *CEP290*-LCA IRD, the pathways modulated by reserpine were enriched in proteostasis-related processes. This study reaffirms the potential of reserpine as a mutation-independent small molecule drug for potential treatment of IRDs and underscores the validity of our screening platform for identifying drug candidates that promote photoreceptor cell survival.

## Results

### Design of study for reserpine treatment of rhodopsin P23H rat model of adRP

Retinal morphological and functional changes in the P23H-1 rats have been widely described (*Fernández-Sánchez et al., 2011*; *LaVail et al., 2018*; *Lu et al., 2013*). Before initiating the study, we evaluated the progression of retinal degeneration in P23H-1 rats at the EyeCRO facility (Oklahoma City, OK). As early as P36, P23H-1 rats exhibited a reduction in the amplitude of scotopic a- and b-waves compared to wild-type animals (*Figure 1—figure supplement 1*). The decline in retinal function continued progressively through P57, at which point the remaining function was comparable to that observed at P156. The first intervention was administered before functional impairment peaked with a goal to harness the remaining potential for rescue and to promote functional improvement. Studies on myopia using chicken eyes previously showed the effects of intravitreally injected reserpine could persist for more than 18 days after the injection (*Ohngemach et al., 1997*; *Schaeffel et al., 1995*). Our pharmacokinetic study in rat eye with another compound of similar molecular weight had shown that intravitreally injected drug could persist at concentration higher than 2 ng/mL throughout the entire observed period, up to 7 days (data not shown). Given the progression of retinal degeneration and duration of the drug effect, bilateral intravitreal injections of 5 µL of either vehicle or 40 µM reserpine were performed at P30 and P44. Visual function was evaluated using electroretinography (ERG), and optokinetic tracking (OKT) to quantify spatial frequency threshold and contrast threshold (CT). Structural changes were assessed using optical coherence tomography (OCT) imaging 10 and 24 days after the second injection (*Figure 1A*). Eyes were enucleated at P70 for histological and molecular analysis.

### Reserpine treatment attenuates the decrease of scotopic b-wave in P23H-1 rats

Two intravitreal injections of either vehicle or reserpine did not result in corneal or lens opacity, nor did they cause observable retinal inflammation in animals. When comparing the vehicle-injected control group with age-matched wild-type rats, ERG analysis showed a decrease in amplitude of rod-derived scotopic response at P54, a further reduction at P68, and a decrease in amplitude of cone-derived photopic response at P68 (*Figure 1B*, *Figure 1—figure supplement 2A*). Among various measurements comparing the reserpine-treated group with controls, only the scotopic b-wave at P68 revealed a statistically significant increase in amplitude in rats treated with reserpine compared to controls (*Figure 1B*). The amplitude was 388.6±136.1 µV in reserpine-treated group, while it was 277.8±62.4 µV in control group (p<0.05). Scotopic a-wave and photopic b-wave amplitude in reserpine-treated group were higher compared to the control group (38.01±14.94 µV vs. 28.60±7.64 µV and 142.8±52.62 µV vs. 104.9±22.12 µV, respectively), but were not statistically significant. Other parameters such as OKT, contrast threshold, and ONL thickness evaluated by OCT did not show significant differences (*Figure 1C*, *Figure 1—figure supplement 2B*).

### Female rats primarily account for the responses to reserpine treatment

Distinct patterns in the scotopic b-wave emerged in the reserpine-treated group when males and females were examined separately (*Figure 2A*, *Figure 2—figure supplement 1*). As shown in

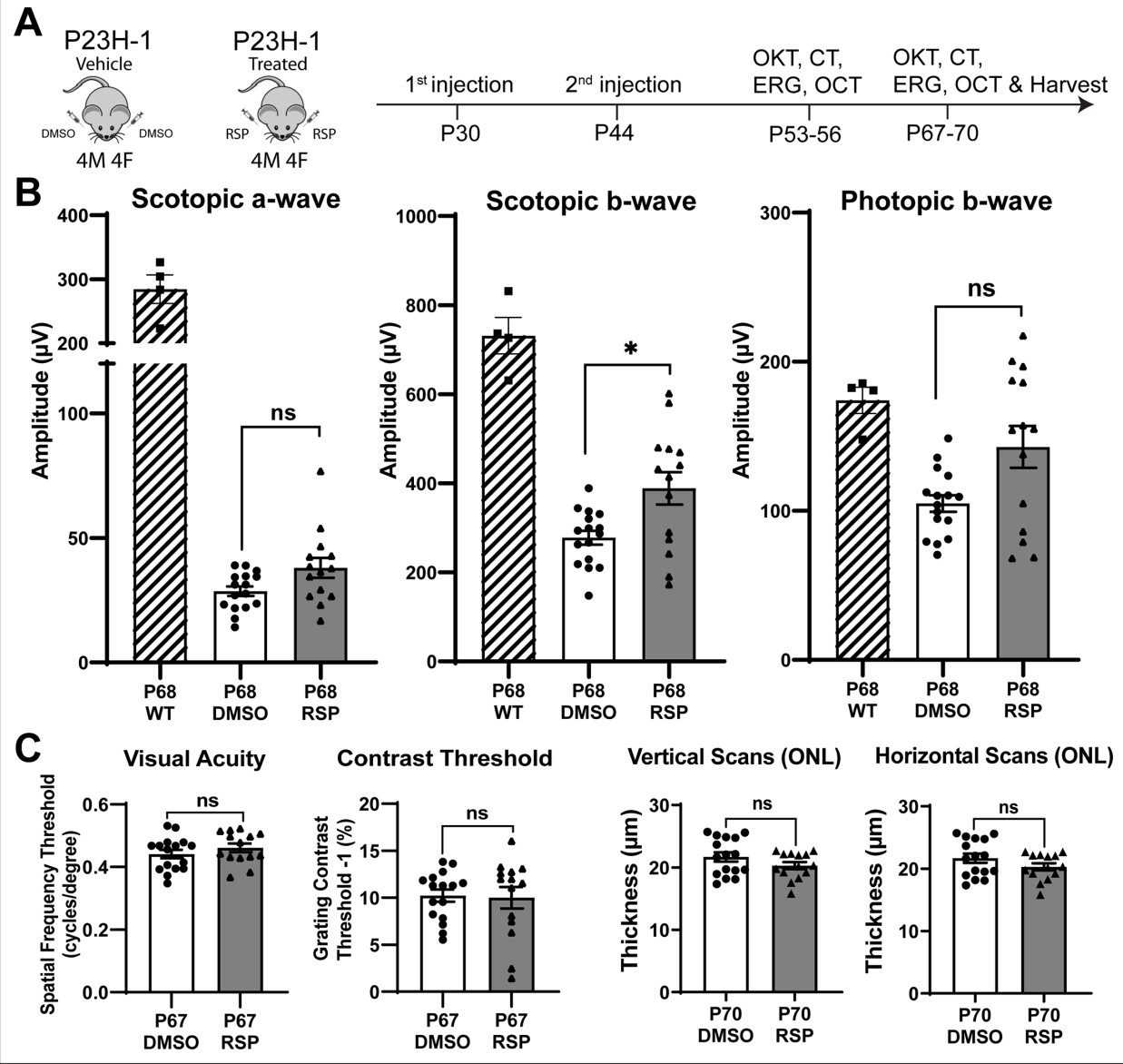

**Figure 1.** Experimental design and treatment responses. (**A**) Timeline of intravitreal injections and functional and structural analysis. (**B**) Scotopic and photopic electroretinogram responses at P68 in wild type, DMSO- and RSP-treated P23H-rats. (**C**) Visual acuity measured by OKT response and contrast threshold, and outer nuclear layer thickness measured by vertical and horizontal scan of OCT in DMSO- and RSP-treated P23H-rats. All parameters were measured in both eyes (8 DMSO-treated and 7 RSP-treated rats). Data were expressed as mean ± SEM, and the Mann-Whitney U test was used to compare DMSO- and RSP-treated groups. RSP: reserpine, OKT: Optokinetic tracking, CT: Contrast threshold, ERG: electroretinogram, OCT: Optical coherence tomography, ONL: outer nuclear layer, ns: not significant, *p<0.05.

The online version of this article includes the following source data and figure supplement(s) for figure 1:

**Source data 1.** Individual values of ERG, OKT responses, and contrast threshold in DMSO- and RSP-treated rats.

**Figure supplement 1.** Progression of retinal degeneration in P23H-1 rats evaluated by electroretinography.

**Figure supplement 2.** Treatment responses in P23H-1 rats.

*Figure 2A*, most individuals with higher amplitude in scotopic b-wave were female rats whereas no difference was evident in male and female rats in control group. We identified the segregation stemmed from differences in responses between sexes through a subgroup analysis of males and females. Female rats also showed slower progression in decrease of scotopic b-wave response between P54 and P68 (*Figure 2B*). When subgroup analysis of female rats was performed, female

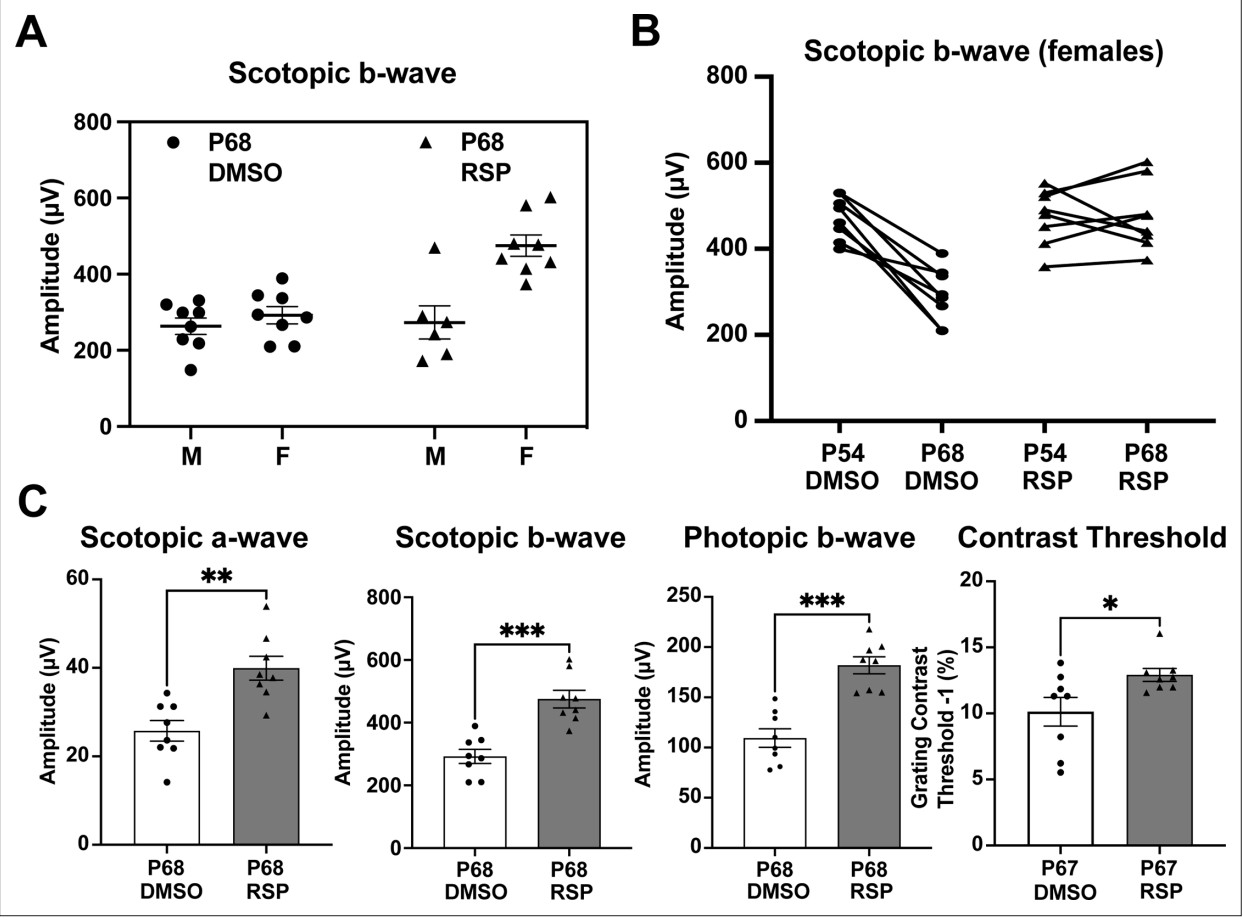

**Figure 2.** Sex-specific treatment responses in ERG analysis. (**A**) Sex-associated subgroup analysis for scotopic b-wave amplitude at P68. (**B**) Individual changes of scotopic b-wave amplitude between P54 and P68 in female rats. (**C**) Scotopic and photopic electroretinogram responses and contrast threshold at P68 in DMSO- and RSP-treated female P23H-rats. Data were expressed as mean ± SEM, and the Mann-Whitney U test was used to compare DMSO- and RSP-treated groups. RSP: reserpine, *p<0.05; **p<0.01; ***p<0.001.

The online version of this article includes the following figure supplement(s) for figure 2:

**Figure supplement 1.** ERG responses in P23H-1 rats grouped by sex.

rats showed higher response in scotopic a- and b-wave, photopic b-wave at P68 and better contrast threshold at P67 in reserpine-treated group (*Figure 2C*).

## Dorsal and temporal retina exhibit prominent structural improvement in female rats

Retinal degeneration in P23H-1 rats does not progress synchronously throughout the retina, but dorsal retina is affected more severely (*García-Ayuso et al., 2013*; *LaVail et al., 2018*). P23H mutations in *RHO* gene mainly affects rod photoreceptor cells that constitute more than 97.5% of photoreceptor cells in rat retina (*Szél et al., 1992*). Therefore, rod photoreceptor degeneration in P23H-1 rats can be evaluated by measuring outer nuclear layer thickness, where photoreceptor nuclei lie. The OCT evaluation of female rats at P70 showed thinner outer nuclear layer (ONL) in the dorsal retina (*Figure 3A*). Treatment with reserpine could preserve the ONL thickness of the dorsal retina at 1,000 **µ**m distant from the optic nerve head, which was 17.93±0.96 **µ**m compared to 15.29±0.73 **µ**m in the control group. Horizontal scan revealed increased ONL thickness of temporal retina at 750 **µ**m distant from the optic nerve in the reserpine-treated group (*Figure 3B*). Consistent with the OCT findings, histologic evaluation of the control group also demonstrated more pronounced ONL thinning in the dorsal retina. Reserpine treatment increased the ONL thickness in the dorsal retina at 1000, 1250, and 1500 µm from the optic nerve head (*Figure 3C*). The number of nuclei in the ONL in SD rats is

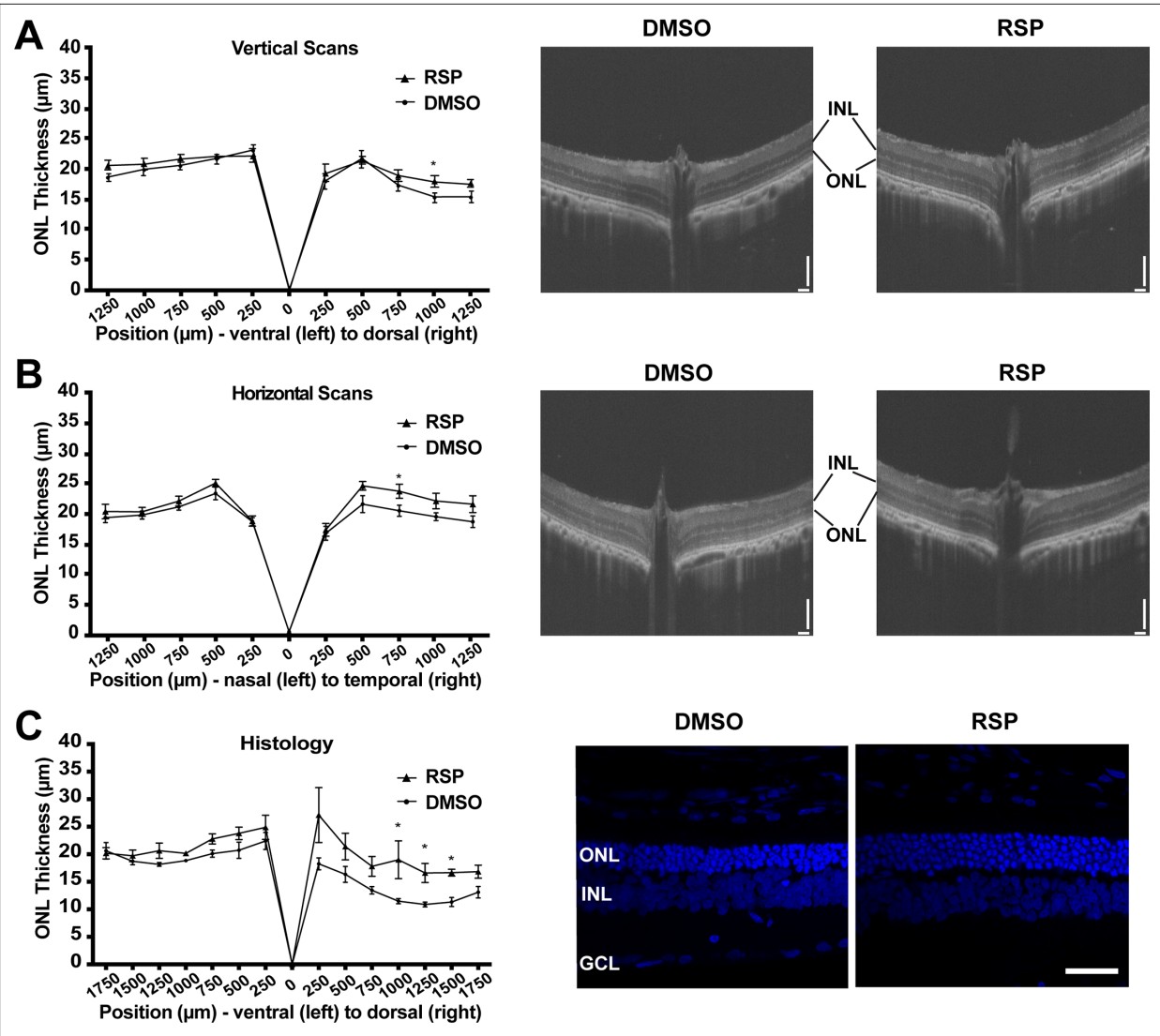

**Figure 3.** Effect of reserpine treatment on outer nuclear layer thickness of female P23H rats. (**A**) Outer nuclear layer thickness evaluated by vertical scan of OCT at P70. (**B**) Outer nuclear layer thickness evaluated by horizontal scan of OCT at P70. (**C**) Outer nuclear layer thickness evaluated by DAPI-stained retinal section at P70. The representative images were taken from dorsal retina, 1000 μm away from the optic nerve. Data were expressed as mean ± SEM, and the Mann-Whitney U test was used to compare DMSO- and RSP-treated groups. Images are representative of four female rats. The scale bar represents 100 μm in A and B, and 20 μm in C. RSP: reserpine, ONL: outer nuclear layer,, INL: inner nuclear layer,, GCL: ganglion cell layer, *p<0.05.

10–14 (*García-Ayuso et al., 2013*), but it was 3–4 nuclei in vehicle treated P23H-1 rats and 5–6 nuclei in reserpine-treated P23H-1 rats.

### Reserpine maintains rod and cone photoreceptor survival in female rats

As reserpine consistently improved retinal function and the number of photoreceptors in P23H-1 female rats, all further analysis was done for female rats only. To evaluate the structure of photoreceptors, retina sections were immunostained with REEP6, rhodopsin and cone arrestin. In P23H-1 rats, early changes such as increase in pyknotic nuclei can be detected as early as at P10 in the dorsal retina, which is followed by progressive loss of photoreceptor nuclei and shortening of rod inner and outer segments (*LaVail et al., 2018*). Immunostaining with REEP6 revealed extensive shortening of rod inner segments in the dorsal retina of vehicle treated rats at P70. In contrast, the inner segments were relatively preserved in the dorsal retina of female rats treated with reserpine (*Figure 4A*). As retinal degeneration progresses in P23H-1 rats, the progressive loss of photoreceptors is accompanied by shortening of the outer segments and mislocalization of rhodopsin to the inner segment

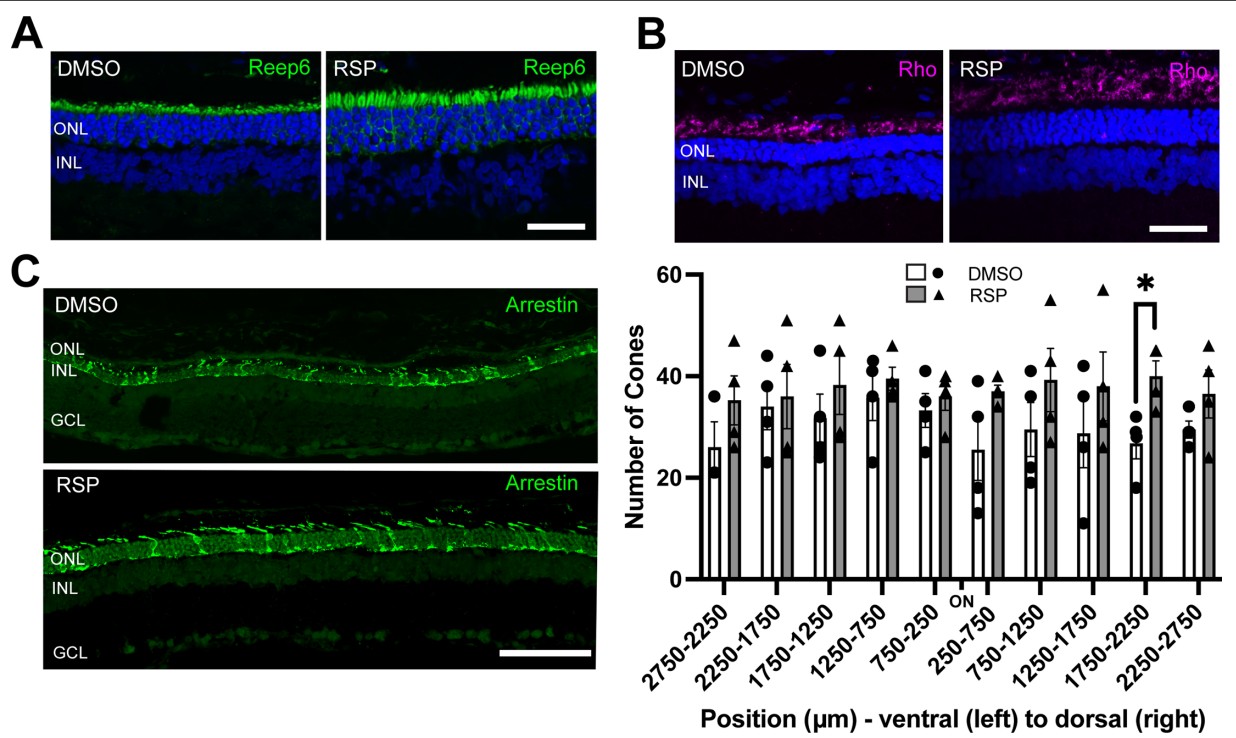

**Figure 4.** Effect of reserpine treatment on retinal photoreceptors of female P23H rats. (**A, B**) Rod photoreceptors were evaluated by immunostaining of REEP6 and rhodopsin (Rho) at P70. The representative images were taken from dorsal retina, 1000 μm away from the optic nerve. The scale bar represents 20 μm. (**C**) Cone photoreceptors were evaluated by immunostaining of cone arrestin at P70. Number of cone photoreceptors were counted every 500 μm throughout the retina. Data were expressed as mean ± SEM, and the Mann-Whitney U test was used to compare DMSO- and RSP-treated groups. Images are representative of four female rats. The scale bar represents 20 μm. RSP: reserpine, ONL: outer nuclear layer, INL: inner nuclear layer, GCL: ganglion cell layer, *p<0.05.

region. In the reserpine-treated female retina, the outer segments were longer, and rhodopsin was located farther from the nuclei compared to the vehicle-treated female retina (*Figure 4B*). In RP, cone photoreceptors gradually degenerate after rod photoreceptor loss (*Campochiaro and Mir, 2018*), and preserving cone photoreceptors can be particularly beneficial for patients in advanced stages of the disease. When cone photoreceptors were counted every 500 μm across the retina, the reserpine-treated female retina showed a higher number of cones at approximately 2000 μm dorsal to the optic nerve head and relatively preserved cone morphology (*Figure 4C*).

## Cellular pathways show sex-biased dysregulation in rhodopsin P23H rat retina

To understand the processes behind reserpine-mediated improvement in photoreceptor survival, we first characterized the molecular pathogenesis of P23H-induced retinal neurodegeneration. We profiled retinal transcriptomes from female and male rats with wild type (WT) and $Rho^{P23H/+}$ genetic backgrounds at P70. For the mutant animals, we collected retinas from both reserpine-treated and untreated control groups. After standard quality check and filtering, a total of 10,726 genes were captured across 19 samples. Principal component analysis (PCA) showed distinct clustering of WT and P23H transcriptomes (*Figure 5A*). Interestingly, female and male retinas separated within WT as well as treated and untreated groups, suggesting a notable influence of biological sex. Therefore, we designed our transcriptomic analyses to test differences when both sexes were combined as well as to check for female and male specific trends.

Combined analyses of both male and female WT and P23H retinas revealed 652 genes differentially expressed at significance threshold of adjusted p-value <0.05 and absolute fold change >2 (*Figure 5—source data 1*). These genes showed strong differences between WT and mutant retinas as well as some biological sex dependent trends within each genotype (*Figure 5B*). These genes summarize

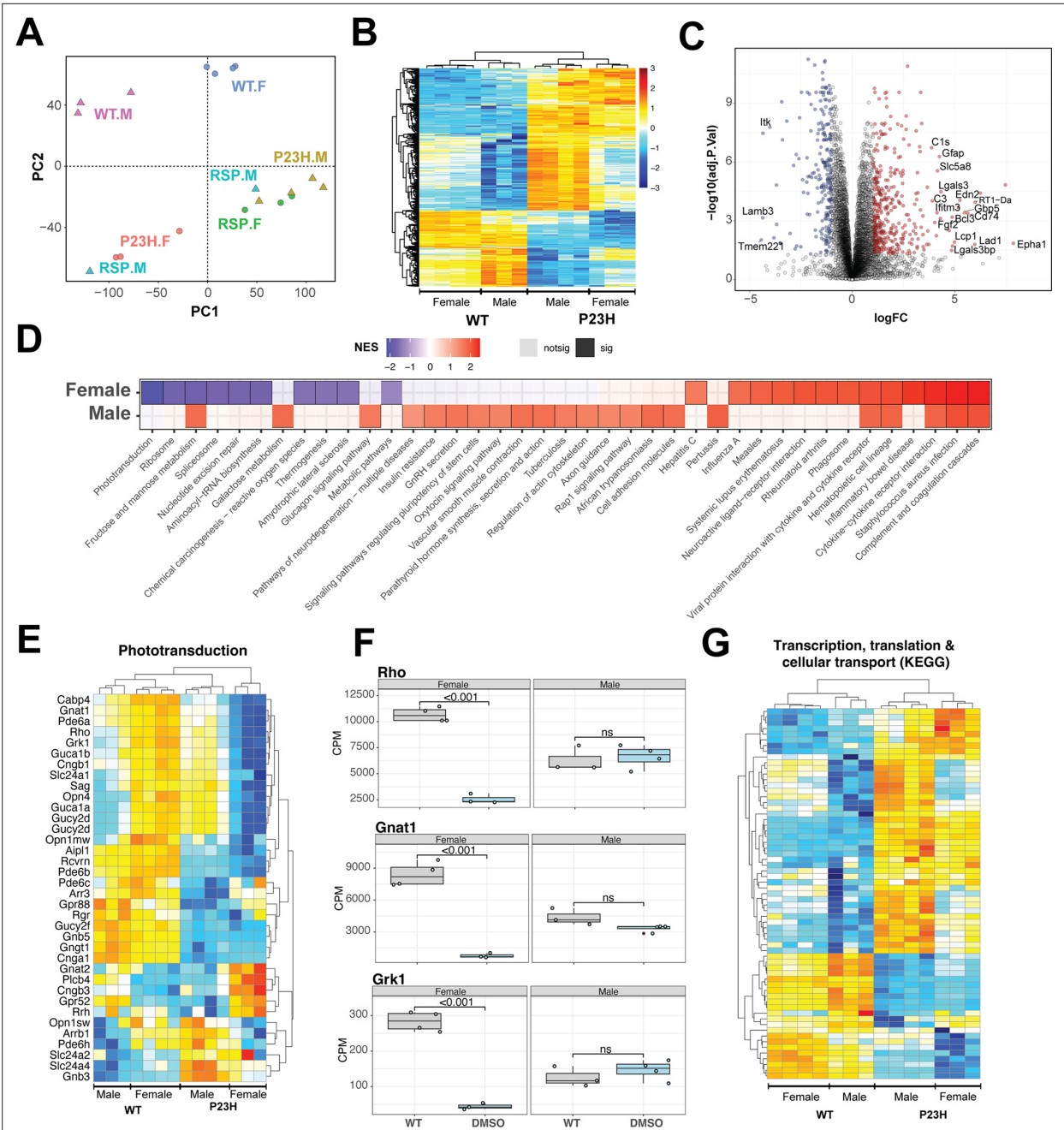

**Figure 5.** Pathology of $Rho^{P23H/+}$ retina impacts key retinal pathways and is influenced by biological sex. (**A**) Principal Component Analysis (PCA) plot of retinal transcriptomes from WT, P23H and P23H reserpine-treated rats at P70. Female (.F) and male (.M) animals are represented by circles and triangles, respectively. (**B**) Heatmap of significantly differentially expressing genes between WT and P23H retinas. Male and female retinas have common as well as unique transcriptomic trends in P23H rodents. Colors are z-scores of row-scaled log2 CPM values of genes. (**C**) Volcano plot of significantly differential genes between WT and P23H retinal transcriptomes. Top over- and under-expressing genes labeled on the plot. (**D**) KEGG pathways enriched in mutant retinal transcriptomes of female and male rats. Over and under-enriched pathways are shown in red and blue, while non-significant enrichment is depicted in transparent fill. (**E**) Phototransduction genes express differently in P23H mutant retinas and have a sex-biased trend. (**F**) Boxplots showing expression of key phototransduction genes – rhodopsin (*Rho*), transducin (*Gnat1*), and rhodopsin kinase (*Grk1*) in female and male retinas. Adjusted p-value from statistical comparison is shown on the plot, with 'ns' denoting a value greater than 0.05. (**G**) Heatmap of significantly differential genes in KEGG pathways of transcription, translation and cellular transport that constitute the genetic information processing supergroup. Heatmap colors in panels (**E**) and (**G**) are z-scores of row-scaled log2 CPM values of genes and similar to the plot in (**B**). WT: wild-type, P23H: DMSO-treated P23H retinas, RSP: reserpine-treated P23H retinas.

The online version of this article includes the following source data and figure supplement(s) for figure 5:

*Figure 5 continued on next page*

*Figure 5 continued*

**Source data 1.** Differential gene expression between WT and P23H retinas.

**Figure supplement 1.** Rho P23H linked molecular pathology is consistent across different model systems.

the transcriptomic signature of P23H pathology and include a host of protein-coding-genes that function in diverse sets of pathways (*Figure 5C*, *Figure 5—figure supplement 1A*). We used gene set enrichment analysis using KEGG pathways to identify positively and negatively enriched processes. Phototransduction was observed to be strongly downregulated in P23H retinas with several genes showing marked sex-specific trends in the mutant (*Figure 5D and E*). We plotted expression of genes for rhodopsin (*Rho*), transducin (*Gnat1*) and rhodopsin kinase (*Grk1*) in female and male retinas to visualize the disease- and sex-differences at individual genes (*Figure 5F*). Many housekeeping and inflammation related gene sets were significantly impacted by P23H-linked transcriptomic changes. We analyzed two published transcriptome datasets from P23H and WT rodent models to evaluate our observations and identified that many of the pathways were enriched in those studies (*Leinonen et al., 2020*; *Vats et al., 2022*; *Figure 5—figure supplement 1B*). For instance, phagosome and phototransduction follow similar trends in all three studies while many others present in more than one dataset indicating their importance in the molecular pathology of P23H linked retinal degeneration. Among the prominent pathways, those relating to transcription, translation and cellular transport that deal with production, quality control and transport of biomolecules were markedly altered in P23H mutant retinas (*Figure 5G*, *Figure 5—figure supplement 1B*). Leading edge genes for phagosome and lysosomal had net positive fold change of gene expressed in comparisons of the mutant and WT animals (*Figure 5—figure supplement 1C*).

To assess the role of biological sex in P23H molecular pathology, we focused on the sex-specific differential analyses between WT and mutant retinal transcriptomes (*Figure 5—figure supplement 1D*). The female P23H rat retina revealed 548 over-expressed genes whereas 796 genes showed lower expression. Male retinas demonstrated stronger differential trends with 2542 and 1580 over- and under-expressing genes, respectively. We also observed that combining male and female datasets appeared to uncover consistent differential genes that were otherwise missed when individual sexes were compared (*Figure 5—figure supplement 1D*). Geneset enrichment analysis of female and male transcriptomes uncovered similar pathways observed in the combined dataset (*Figure 5D*, *Figure 5—figure supplement 1A*). Female retinas had a wide range of positively and negatively enriched pathways, whereas male transcriptomes primarily presented pathways with positive net enrichment (*Figure 5D*). Our results indicate an interesting variance in P23H pathology in retinas of female and male rats suggesting that the role of biological sex should be investigated in inherited retinal degeneration as well as in the broader context of retinal health and disease.

## RNA-seq identifies sex-specific responses to reserpine treatment

The transcriptome analysis revealed biological sex-related alterations in P23H mutant retinas when comparing to WT as well as drug response in mutants (*Figure 5A*). Following sex-specific molecular pathology trends, we identified differential genes between sex-matched treated and untreated retinal transcriptomes. In female retinas, the reserpine treatment resulted in higher expression of 1437 genes, whereas 352 genes showed lower expression (*Figure 6A*, *Figure 6—source data 1*). Expression profiles of differential genes clustered closer to WT than untreated mutant retinas as evidence by the dendrogram in the heatmap of *Figure 6A*. In contrast, male P23H retinas showed only 180 over and 580 under expressing genes, however, unlike female samples, male differential genes did not cluster with WT suggesting a distinct reserpine response in the two sexes (*Figure 6—figure supplement 1A and B*, *Figure 6—source data 1*). KEGG pathway enrichment (using ClueGO) of overexpressing genes in reserpine-treated females uncovered several key pathways responding to treatment (*Figure 6B*). Most promisingly, we observed elevated expression across phototransduction genes in treated female retinas, including that of the gene coding for rhodopsin (*Figure 6C*). Also responding to reserpine were the key rod photoreceptor transcription factors - NRL, NR2E3, and CRX (*Figure 6D*). Interestingly, MEF2C, another transcription factor associated with rod gene expression (*Hao et al., 2011*) showed an opposite trend of lower expression in reserpine-treated female retinas (*Figure 6—figure supplement 1D*).

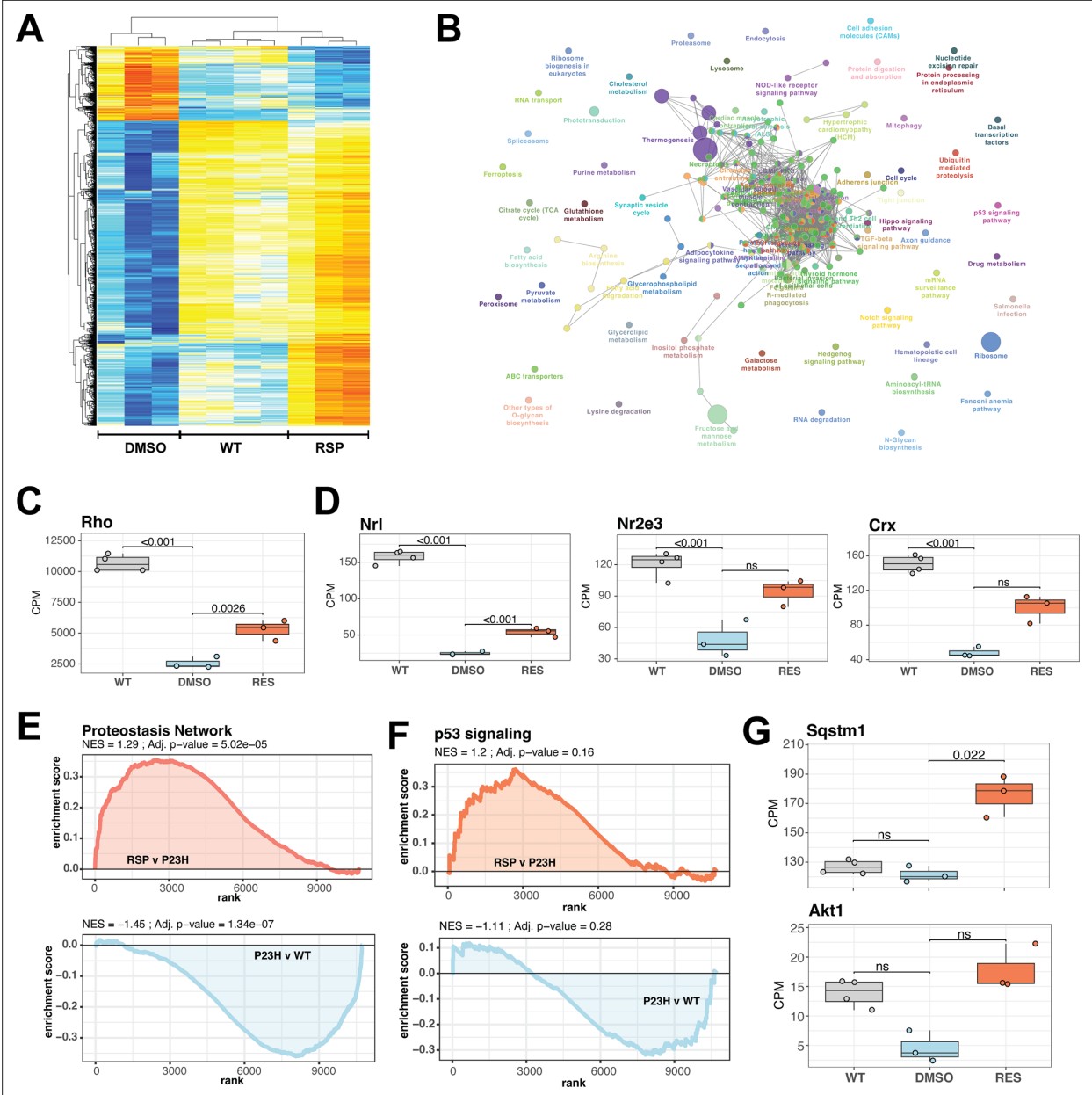

**Figure 6.** Reserpine treatment leads to improved phototransduction and key signaling processes in retinas of female P23H rats. (**A**) Heatmap of reserpine associated differentially expressing genes showing clustering of transcriptomic profiles of treated retinas with that of WT rats. Colors are z-scores of row-scaled log2 CPM values of genes and same as in *Figure 5B*. (**B**) KEGG pathway enrichment network of reserpine linked over-expressing genes highlighting clusters of pathways impacted by treatment. Analysis performed using ClueGo on Cytoscape. Notable pathways include phototransduction, proteostasis network and p53 signaling among others. (**C**) Reserpine-treated female P23H retinas have higher gene expression of Rho than that of DMSO administered samples. (**D**) Gene expression of several key photoreceptor transcription factors – Nrl, Nr2e3, and Crx – recover after reserpine treatment. (**E**) Transcription of proteostasis network genes in female mutant retinas is reversed by reserpine. (**F**) Reserpine leads to transcriptomic upregulation of p53 signaling pathway genes. (**G**) *P62 (Sqstm1)* and *Akt1* genes are transcribed higher in response to reserpine administration. Adjusted p-value from statistical comparison is shown on the plot, with 'ns' denoting a value greater than 0.05. WT: wild-type, P23H and DMSO: DMSO-treated P23H retinas, RSP: reserpine-treated P23H retinas.

The online version of this article includes the following source data and figure supplement(s) for figure 6:

**Source data 1.** Differential gene expression in P23H retinas after reserpine treatment.

**Figure supplement 1.** Distinct transcriptomic response to reserpine in female and male mutant retinas.

We then investigated the expression of cell type-specific markers to estimate impact of reserpine on retinal cell types and observed diverse response; however, reserpine-treated samples clustered closer to WT than untreated P23H female retinas (*Figure 6—figure supplement 1C*). A few photoreceptor markers show increased expression with treatment, as did genes for bipolar, horizontal, amacrine and retinal ganglion cells. Notable drug response was observed in proteostasis linked pathways such as proteasome, endocytosis, lysosome and ubiquitin-mediated proteolysis, which had at least five significantly over-expressing genes and a minimum 5% of the pathways impacted (*Figure 6B*). Furthermore, signaling pathways such as p53, TGF-beta and hippo were also impacted by reserpine linked over-expression. Pathway analysis imply that reserpine may be acting via similar mechanisms described in our previous report of an unrelated retinal pathology and disease model (*Figure 6B*; *Chen et al., 2023*). Therefore, we ran a geneset enrichment analysis of 'Proteostasis Network' and recorded significant enrichment in mutant retinas upon reserpine treatment (*Figure 6E*). Similar strong activation post drug treatment was observed in p53 signaling (*Figure 6F*). To further understand upregulation of proteostasis network, we checked two key factors p62 (Sqstm1) and Nrf2 (Nfe2l2) and identified robust overexpression in the former while the latter did not show significant response despite an increasing trend (*Figure 6G*, *Figure 6—figure supplement 1E*). In addition, several genes for proteasomal regulation were significantly differentially expression in reserpine-treated female P23H retinas (*Figure 6—figure supplement 1F*). Similarly, the gene coding for Akt1, a prominent pro-survival serine/threonine kinase and a member of the p53 network, was significantly responsive to reserpine, as were other genes of the pathway (*Figure 6G*, *Figure 6—figure supplement 1G*). Finally, KEGG neuronal system genesets including multiple synapse related processes were observed to show higher gene expression in female P23H retinas upon reserpine treatment (*Figure 6—figure supplement 1H*).

## Discussion

Vision-impairing retinal degenerative diseases are debilitating, with significant burden on patients, families and the society. Notable progress has been made in cataloging IRD genes and mutations, understanding molecular pathogenesis, as well as translational innovations for retinal diseases (*Duncan et al., 2018*; *Garafalo et al., 2020*; *Jiang et al., 2022*). Initial applications of gene replacement or gene augmentation have shown promising outcomes (*Drag et al., 2023*; *Kruczek et al., 2021*). However, the imposing diversity of IRDs with hundreds of disease genes and thousands of mutations dramatically impact the prospect of therapies and their implementation worldwide. As a result, gene-independent approaches, where disease adjacent processes are targeted for alleviating or circumventing the influence of a genetic mutation, are attractive and finally gaining widespread appreciation in vision research. Such therapies hold great promise for late-onset or slowly progressing IRDs, providing a valuable opportunity for intervention and potentially mitigating neurodegeneration before a comprehensive cure is available. For instance, pharmacological intervention of rhodopsin-linked disease pathways, such as the unfolded protein response or endoplasmic reticulum stress, among others (*Zhen et al., 2023*), can be predicted to moderate rod degeneration associated with the numerous autosomal dominant or recessive mutations in RHO. Similarly, normalization of metabolic imbalance may be able to alter the trajectory of retinal degeneration (*Hurley et al., 2015*; *Nolan et al., 2022*; *Xue et al., 2024*). Gene-agnostic treatment in this paradigm would focus on survival of rod photoreceptors for improving cone survival and protection of high acuity vision (*Curcio et al., 2000*). Previously, we demonstrated enhanced photoreceptor survival in *CEP290*-related retinal ciliopathy by a repurposed drug, reserpine (*Chen et al., 2023*). Our findings, reported here, show the potential of reserpine as a gene-independent drug for IRDs, especially those involving proteostasis defects. We also uncover, by transcriptome analysis, additional pathways influenced by reserpine. Thus, photoreceptor survival upon reserpine treatment appears to be an outcome of realignment among multiple disease gene-independent pathways.

Reserpine-treated P23H retina exhibits improved visual function, with higher and consistent potency in female rats, where scotopic a- and b-wave and photopic b-wave are better preserved. Structural analysis with OCT and histology also reveals prominent response in dorsal regions of reserpine-treated retina where degeneration is known to be severe and linked with light exposure (*Bogéa et al., 2015*; *García-Ayuso et al., 2013*; *LaVail et al., 2018*; *Tam et al., 2010*). Thus, reserpine is effective in maintaining photoreceptor survival in rodent models of an LCA as well as a RP mutation.

Notably, the photoreceptor response in P23H rat model was measured three weeks after the last reserpine injection, showing the longer-term effect of the treatment. Retinal transcriptome analysis reveals alterations in proteostasis network, as well as in P53 and AKT signaling, among other pathways to concur with photoreceptor survival in reserpine-treated P23H retinas. Remarkably, some of the drug-responding pathways are reportedly dysregulated in another retinal degeneration model (*Jiang et al., 2022*). Further investigations are warranted in model systems with different IRD mutations to evaluate the impact of reserpine treatment and to elucidate pro-disease and pro-survival pathways that will support future developments of gene-agnostic therapies.

Reserpine, originally FDA-approved for hypertension treatment, inhibits vesicular monoamine transporter (VMAT), depleting catecholamines by preventing their storage and leading to their metabolism by monoamine oxidase (*Koch et al., 2020*). Intravitreal reserpine injections in chickens depleted retinal dopamine and serotonin, with effects lasting up to 21 days (*Ohngemach et al., 1997*; *Schaeffel et al., 1995*). However, ERG measurements and spectral sensitivity showed no differences in normal retinas treated with reserpine (*Schaeffel et al., 1995*). Rodent photoreceptors express $D_2$-like receptors (*Derouiche and Asan, 1999*; *Patel et al., 2003*; *Tran and Dickman, 1992*) and $5\text{-}HT_{3A}$ receptors (*Masson, 2019*), with $D_2$-like receptor antagonists reducing rod response amplitude and kinetics, whereas agonists enhance the response (*Jin et al., 2015*). Reserpine-induced catechol-amine depletion likely reduces the scotopic b-wave amplitude at P54 (*Figure 1—figure supplement 2A*); however, no further decrease is detected after that, unlike a progressive reduction in untreated controls. The effects of catecholamine receptor modulators on retinal disease may vary depending on experimental conditions. Antagonists for $D_1$-, $D_2$-like or $5\text{-}HT_{2A}$ receptors, as well as dopamine deple-tion, have shown protective effects (*Ogilvie and Speck, 2002*; *Tullis et al., 2015*), and agonists for $D_2$-like or $5\text{-}HT_{1A}$ receptors have also demonstrated protective effects in different contexts (*Collier et al., 2011*; *Tanaka et al., 2021*). Combination therapies targeting adrenergic, dopamine, and serotonin receptors appear to show promise for retinal protection in diabetic retinopathy and inherited retinal degeneration (*Kern et al., 2021*; *Leinonen et al., 2024*). Furthermore, reserpine-linked activation of proteostatic mechanisms and vital signaling processes highlight broader impact of the drug when tested in vivo.

Our study highlights a distinct sex-bias in disease profiles as well as response to reserpine in female and male P23H rats, where female rats exhibit significant recovery of retinal function as measured by ERG. Transcriptomic comparisons among WT, P23H mutant, and reserpine-treated mutant retinas also showed distinct trends in female and male rats. Though retinal transcriptomes from P23H retinas from both sexes deviate significantly from WT, many disease-linked enriched pathways are unique to a sex group. Treated female retinas have significantly higher expression of the gene for P62 (SQSTM1), indicating at a potential key route for reserpine's activity. Furthermore, recovery in gene expression of important retinal transcription factors hint at a consequential influence of reserpine's gene-agnostic impact. Our previous study demonstrated that retinal organoids derived from female LCA10 patients are more responsive to reserpine, while male-derived organoids show less consistent responses (*Chen et al., 2023*). Overall, this exposes the multifaceted nature of IRDs in general and advocates inclusion of biological sex as a study variable in ophthalmic research (*Arnegard et al., 2020*) in line with NIH policy.

Sex differences in response to reserpine have been widely observed. When striatal dopamine storage was impaired by the inhibition of VMAT by reserpine, a greater degree of dopamine deficits was observed in female mice, suggesting that females possess a greater dopamine storage capacity (*Ji et al., 2007*). The greater depletions of striatal dopamine content in female mice was not estrogen dependent (*Dluzen et al., 2008*). Despite the increased responsiveness of dopamine levels in females, another study showed that female rats were resistant to cognitive and motor deficits in reserpine-induced Parkinson's disease model (*Lima et al., 2021*). In contrast, Rivera-Garcia et al. suggested that some sex differences are strain-dependent, showing no significant difference in reserpine-evoked dopamine release between female and male Long-Evans rats, but observing sex differences in elec-trically stimulated dopamine release in Sprague-Dawley rats (*Rivera-Garcia et al., 2020*). Beyond the response to reserpine, ERG responses without any treatment may also exhibit sex differences. For instance, 18-month-old female rats showed ERG changes according to menopausal status, whereas no significant differences in ERG or retinal histology were observed between male and female rats at any other age (*Chaychi et al., 2015*).

Our study further illustrates the potential of reserpine as a gene-independent therapeutic agent by extending photoreceptor survival in distinct models of IRDs. We extend our published findings on *CEP290*-LCA models (*Chen et al., 2023*). Employing a range of molecular, histological and functional assays, we demonstrate the benefits of in vivo reserpine application in a popular rodent model of adRP. We elucidate possible drug response mechanisms and uncover a major influence of biological sex. However, the molecular target of reserpine in the retina remains unknown although we note possibilities of a different associate in the retina. Additionally, reasons for biological sex bias in drug response must be investigated with a focus on determining whether female and male retinas are predisposed differently to retinal pathology and interventions. Overall, our findings highlight the broader application of small molecules derived from unbiased large-scale screening and underscore the importance of considering sex-specific differences in the development of treatments for IRDs.

# Materials and methods

**Key resources table**

| Reagent type (species) or resource | Designation | Source or reference | Identifiers | Additional information |
|---|---|---|---|---|
| Antibody | Anti-REEP6 (rabbit polyclonal) | PMID:28369466 | Available from Dr. Swaroop | IF(1:1000) |
| Antibody | Anti-RHO (mouse monoclonal) | Gift of Dr. Robert Molday, University of British Columbia | Please request it from Dr. Molday | IF(1:500) |
| Antibody | Anti-cone arrestin (rabbit polyclonal) | Millipore | Cat# AB15282, RRID:AB_1163387 | IF(1:500) |
| Chemical compound, drug | Reserpine | Sigma-Aldrich | 06859 | |
| Software, algorithm | ImageJ | ImageJ | RRID:SCR_003070 | https://imagej.net/ij/ |
| Software, algorithm | Photoshop 2023 | Adobe | RRID:SCR_014199 | https://www.adobe.com/products/photoshop.html |
| Software, algorithm | Biowulf Linux cluster | National Institutes of Health | RRID:SCR_007169 | http://biowulf.nih.gov |
| Other | DAPI stain | Invitrogen | Cat# D1306, RRID:AB_2629482 | (1 µg/mL) |

## Animals

Hemizygous P23H-1 rats were produced by crossing homozygous transgenic P23H line 1 rats, obtained from Matthew LaVail (University of California, San Francisco School of Medicine; *LaVail et al., 2018*), with wild-type Long Evans rats purchased from Charles River. The hemizygous P23H-1 rats carry a single P23H transgene allele with a transgene copy number of 9, in addition to the normal two wild-type opsin alleles (*Orhan et al., 2015*). The ratio of transgene to endogenous opsin mRNA was 0.1:1 (*LaVail et al., 2018*; *Machida et al., 2000*). For control animals, wild-type Sprague-Dawley rats were crossed with Long-Evans rats, also from Charles River. Seven to eight animals per were used per experimental group, and each was assigned a five-digit ID number via an ear tag for tracking purposes. Rats were housed in groups of 3–5 in spacious cages situated on ventilated shelves, following a 12 hr dark/light cycle. Each cage was equipped with refuges, gnawing, and nesting material. Daily monitoring was conducted to detect gross behavioral changes such as lack of activity, lack of grooming, and intolerance, which could be considered humane end points. All procedures were carried out in accordance with guidelines approved by the Institutional Animal Care and Use Committee of EyeCRO (2021-10-16-001).

## Intravitreal injection

Prior to each procedure, pupils were dilated, and a combination of ketamine (85 mg/kg) and xylazine (4 mg/kg) was administered via intraperitoneal injection, following light sedation using isoflurane. A total volume of 5 µL of either 0.4% DMSO or 40 µM reserpine (diluted from 10 mM stock solution) was injected into the vitreous at the pars plana using a Hamilton syringe with a 33-gauge needle. Upon completion of the procedures, Yohimbine was administered via intraperitoneal injection at a concentration of 1 µL per gram of body weight as a reversing agent for ketamine-xylazine anesthesia. Eight

male and eight female rats from two litters were randomly allocated to two groups, with each group receiving bilateral injections of either vehicle or reserpine. Information about the injected materials was blinded to the researchers who conducted the animal experiments and performed the initial analysis for each experiment. One male rat from reserpine-treated group was excluded from downstream analysis due to death during anesthesia at P53.

## Optokinetic tracking (OKT)

OKT response was measured using an OptoMotry designed for rodent use (Cerebral Mechanics Inc) (*Prusky et al., 2004*). After rats were positioned on a platform surrounded by four LCD screens in a light-protected box, visual stimuli were given to the rats. The LCD monitors in the box displayed continuous vertical sine wave gratings that rotates at a rate of 12 degrees per second, creating the perception of a virtual three-dimensional rotating sphere for the rats. The rotation of the virtual cylinder was adjusted to keep a viewing distance consistent. Using a digital camcorder mounted on the top of the box, a masked observer evaluated the OKT responses that could be identified as slow and steady head movements in the direction of the rotating grating. At a range of spatial frequencies from 0.064 to 0.514 cycles/degree, input from the masked observer was given to the device and the testing stimuli was automatically adjusted based upon the tracking reflex of the animals to identify and determine to spatial frequency threshold which elicited which tracking behavior for each eye. To determine contrast threshold, a fixed spatial frequency threshold of 0.064 cycles/degree was utilized with a decreasing contrast grating stimulus presented to the eyes. The contrast threshold was calculated as a reciprocal of the Michelson contrast from the screen's luminance (maximum – minimum)/(maximum +minimum).

## ERG recordings

After a minimum of 12 hr dark adaptation, rats were anesthetized with an intraperitoneal injection of ketamine (85 mg/kg) and xylazine (4 mg/kg) after pupil dilation. The Espion System from Diagnosys LLC (Lowell, Massachusetts, USA) was employed for both stimulation and recordings. First, a stimulus intensity of 40 (S) cd·s/m$^2$ was presented to record scotopic responses. The amplitude of the scotopic a-wave was measured from the baseline to the trough of the a-wave, while the amplitude of the scotopic b-wave was measured from the trough of the scotopic a-wave to the crest of the scotopic b-wave. Rats were then light adapted for 7 min and were presented with a total of 15 repeated flashes at an intensity of 10 (S) cd·s/m$^2$. The recordings were averaged to produce the final waveform. The amplitude of the photopic a-wave was not analyzed due to its low amplitude and high variability, which is consistent with findings from a previous study (*Orhan et al., 2015*), whereas the amplitude of the photopic b-wave was measured from the trough of the a-wave to the crest of the b-wave.

## Fundus imaging and optical coherence tomography

Following sedation and pupil dilation, rats were securely positioned on an animal stand and fundus was imaged with Micron IV (Phoenix Research Inc, Pleasanton, CA). With Micron IV OCT module, OCT images crossing the optic disc vertically and horizontally were acquired by averaging 25 scans each to eliminate artifacts. After curated layer segmentation, the total retina thickness was evaluated by measuring distance between inner limiting membrane (ILM) and retinal pigment epithelium (RPE), and the outer nuclear layer (ONL) thickness was evaluated by measuring distance between outer plexiform layer (OPL) and RPE. After adjustment of the center, the thickness was obtained at five lateral distances from the optic disc (250, 500, 750, 1000, 1250 µm).

## Immunohistochemistry

Following sedation, animals were euthanized by intracardial administration of Euthasol (Virbac, Westlake, TX). The superior portion of each eye was then demarcated with a scorch mark, and the eyes were enucleated keeping optic nerve intact. For the right eyes, retinas were individually dissected, snap-frozen, and stored at –80 °C until RNA-seq experiments. The left eyes were fixed in 4% PFA for 1 hr at room temperature, then cryoprotected by sequentially incubating in 10% and 20% sucrose-PBS for 1 hr at room temperature, followed by overnight incubation in 30% sucrose-PBS at 4 °C. After removing the cornea and lens, eyes were cryo-embedded in optimal cutting temperature medium (Sakura Finetek, Torrance, CA). Retinas were sectioned vertically at 12 µm on a cryostat and washed

twice in PBS. After blocking with 5% normal donkey serum in 0.5% Triton X-100 dissolved in filtered PBS (PBS-T) for 1 hr at room temperature, slides were incubated overnight at 4 °C with the following primary antibodies: a custom rabbit polyclonal antibody to REEP6 (1:1000, *Veleri et al., 2017*), a mouse monoclonal antibody to rhodopsin (1:500, Gift of Dr. Robert Molday, University of British Columbia), or a rabbit polyclonal antibody to cone arrestin antibody (1:500, Millipore, AB15282). Following three washes in PBS, slides were incubated with a corresponding secondary antibody and 1 µg/ml of 4,6-diamidino-2-phenylindole (DAPI) for 1 hr at room temperature. Sections were then washed three times in PBS, mounted in Fluoromount-G mounting medium (SouthernBiotech, Birmingham, AL), and sealed. Using the tile scan function of the Nikon A1R confocal microscope (Nikon, Tokyo, Japan), entire retinal sections were imaged at high resolution. The ONL thickness was measured at every micron along the entire retina using the ThicknessTool (*Maidana et al., 2020*) ImageJ macro on a semi-manually drawn ONL outline. Cones within a designated boundary, defined by distance from the optic nerve, were counted manually.

### RNA extraction and library preparation

Total RNA was purified from homogenized rat retinas (WT, DMSO controls, and reserpine-treated samples) using TriPure isolation reagent (Roche, Indianapolis, IN) according to the manufacturer's protocol. The quality and quantity of RNA were both assessed using the RNA ScreenTape assay on the Agilent TapeStation system. Samples (with RNA integrity number >8) were used for library generation using the TrueSeq RNA sample Prep Kit v2 (Illumina).

### RNA-seq and data analysis

Retinal transcriptomes were profiled using bulk-RNA sequencing with 125 bp paired-end reads using Illumina sequencing and analyzed with pipelines described before (*Chen et al., 2023*; *Mondal et al., 2024*). In brief, quality check and reference mapping were performed with Kallisto using the *Rattus norvegicus* assembly and gene annotation from Ensembl. Alignments were processed using tximport and further analyzed with the edgeR and limma voom pipelines for differential expression analyses for various comparisons of WT, P23H and reserpine-treated samples. When specified, female and male samples were analyzed separately to characterize sex-bias disease mechanism as well as drug response. Significantly differential genes were mapped to KEGG pathways using ClueGO. Additionally, gene set enrichment analysis was performed using fgsea to identify enriched pathways (KEGG) in the various comparisons. Unless otherwise mentioned, the R programming language, base packages and the tidyverse groups of packages were used for analyses and data visualization. We compared our results with two publicly available retinal transcriptome datasets of the P23H model, generated by independent research groups (GSE152474; *Leinonen et al., 2020*) and (GSE179754, *Vats et al., 2022*). For the former, the list of P23H vs WT enriched pathways were downloaded from GEO; and, for the latter read counts for 1DIV samples (in vitro retinal explants from Rho P23H/+and WT animals treated with DMSO for 24 hr) were analyzed using the same pipeline used for our data to identify differential genes and enriched KEGG gene sets.

### Statistical analysis

Based on the normative data from the facility, the sample size of animals used in each group was determined by an unpaired t-test to be more than six rats in each group (http://www.biomath.info). The Mann-Whitney U test, conducted using GraphPad Prism 10 (GraphPad Software, Inc, La Jolla, CA), was used to compare the means between control and reserpine-treated groups for optokinetic tracking, contrast threshold, scotopic a- and b-wave amplitude, photopic b-wave amplitude, outer nuclear thickness (measured by OCT and histology), and the number of cone photoreceptors. All data were expressed as mean ± standard deviation (SD) unless specified. Results with a p-value <0.05 were considered statistically significant.

### Material availability statement

All materials are available upon request. A Material Transfer Agreement is needed to comply with the guidelines and policies of the National Institutes of Health.

## Acknowledgements

We thank members of the Swaroop laboratory for stimulating discussions and the NEI Biological Imaging Core Facility for technical help. Bioinformatic analyses utilized the high-performance computational capabilities of the Biowulf Linux cluster at the National Institutes of Health (http://biowulf.nih.gov).

## Additional information

### Competing interests
Anupam Mondal, Holly Y Chen, Anand Swaroop: Co-inventor on a patent application filed by NIH. "Small Molecule Drug Candidates And Disease-associated Signature Genes For Therapeutic Interventions In Retinal Degeneration." NIH Reference Number: E-071-2020. Phillip Vanlandingham, Rafal Farjo: Affiliated with EyeCRO. The other authors declare that no competing interests exist.

### Funding

| Funder | Grant reference number | Author |
| --- | --- | --- |
| National Eye Institute | Z01EY000450 | Anand Swaroop |
| National Eye Institute | Z01EY000546 | Anand Swaroop |

The funders had no role in study design, data collection and interpretation, or the decision to submit the work for publication.

### Author contributions
Hyun Beom Song, Conceptualization, Formal analysis, Validation, Investigation, Methodology, Writing – original draft, Writing – review and editing; Laura Campello, Formal analysis, Validation, Investigation, Writing – original draft, Writing – review and editing; Anupam Mondal, Data curation, Software, Formal analysis, Visualization, Writing – original draft, Writing – review and editing; Holly Y Chen, Resources, Validation, Writing – review and editing; Milton A English, Resources, Methodology, Writing – review and editing; Michael Glen, Investigation, Writing – review and editing; Phillip Vanlandingham, Formal analysis, Investigation, Methodology, Writing – review and editing; Rafal Farjo, Conceptualization, Resources, Formal analysis, Supervision, Writing – original draft, Project administration, Writing – review and editing; Anand Swaroop, Conceptualization, Resources, Formal analysis, Supervision, Funding acquisition, Writing – original draft, Project administration, Writing – review and editing

### Author ORCIDs
Hyun Beom Song https://orcid.org/0000-0002-3500-2984
Laura Campello https://orcid.org/0000-0002-0869-1315
Anupam Mondal https://orcid.org/0000-0002-3572-6392
Holly Y Chen https://orcid.org/0000-0001-8320-6714
Phillip Vanlandingham https://orcid.org/0000-0003-1884-6525
Anand Swaroop https://orcid.org/0000-0002-1975-1141

### Ethics
All animal procedures were carried out in accordance with guidelines approved by the Institutional Animal Care and Use Committee of EyeCRO (2021-10-16-001).

Reviewer #1 (Public review): https://doi.org/10.7554/eLife.103888.3.sa1
Author response https://doi.org/10.7554/eLife.103888.3.sa2

## Additional files

### Supplementary files
MDAR checklist

## Data availability

High throughput sequencing datasets generated in this study are available through GEO accession # GSE278306 (https://www.ncbi.nlm.nih.gov/geo/query/acc.cgi?acc=GSE278306).

The following dataset was generated:

| Author(s) | Year | Dataset title | Dataset URL | Database and Identifier |
|---|---|---|---|---|
| Song H, Campello L, Mondal AK, Chen HY, English MA, Glen M, Vanlandingham P, Farjo R, Swaroop A | 2024 | Sex-specific attenuation of photoreceptor degeneration by reserpine in a rhodopsin P23H rat model of autosomal dominant retinitis pigmentosa | https://www.ncbi. nlm.nih.gov/geo/ query/acc.cgi?acc= GSE278306 | NCBI Gene Expression Omnibus, GSE278306 |

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
