## [Editor Report · eLife Assessment]

This **important** Research Advance presents **compelling** evidence on the neuroprotective effects of reserpine in a well-established model of retinitis pigmentosa (P23H-1). This study builds on previous work establishing reserpine as a neuroprotectant in models of Leber congenital amaurosis. Here authors show reserpine's disease gene-independent influence on photoreceptor survival and emphasizes the importance of considering biological sex in understanding inherited retinal degeneration and the impact of drug treatments on mutant retinas. The work will be of interest to vision researchers as well as a broad audience in translational research.

---

## [Referee Report · Reviewer #1 (Public review)]

Summary:

The authors investigate the neuroprotective effect of reserpine in a retinitis pigmentosa (P23H-1) model, characterized by a mutation in the rhodopsin gene. Their results reveal that female rats show better preservation of both rod and cone photoreceptors following reserpine treatment compared to males.

Strengths:

This study effectively highlights the neuroprotective potential of reserpine and underscores the value of drug repositioning as a strategy for accelerating the development of effective treatments. The findings are significant for their clinical implications, particularly in demonstrating sex-specific differences in therapeutic response.

Weaknesses:

The main limitation is the lack of precise identification of the specific pathway through which reserpine prevents photoreceptor death.

Comments on revisions:

Thank you for your thorough revisions. I appreciate the effort you have put into addressing all the concerns I previously raised. Upon reviewing your responses and the updated manuscript, I find that you have adequately clarified the issues and incorporated the necessary modifications. Your revisions have strengthened the paper, and I have no further concerns at this stage.

---

## [Author Response]

The following is the authors’ response to the previous reviews

**Reviewer #1 (Public review):**
Summary:The authors investigate the neuroprotective effect of reserpine in a retinitis pigmentosa (P23H-1) model, characterized by a mutation in the rhodopsin gene. Their results reveal that female rats show better preservation of both rod and cone photoreceptors following reserpine treatment compared to males.Strengths:This study effectively highlights the neuroprotective potential of reserpine and underscores the value of drug repositioning as a strategy for accelerating the development of effective treatments. The findings are significant for their clinical implications, particularly in demonstrating sex-specific differences in therapeutic response.

We sincerely appreciate the reviewer’s comments.

Weaknesses:The main limitation is the lack of precise identification of the specific pathway through which reserpine prevents photoreceptor death.

We acknowledge that the exact pathway through which reserpine exerts its protective effects on photoreceptors remains undetermined, yet our findings provide critical insights into potential mechanisms. Together with our previous report [PMID: 36975211], the studies being presented here validate proteostasis (including autophagy) and p53 signaling as the key pathways underlying reserpine-mediated survival of photoreceptors in retinal disease models. We also go a step further by showing an influence of the biological sex.

We emphasize that the primary aim of this study was to demonstrate the effectiveness of reserpine in a different retinal degeneration model—specifically, the autosomal dominant RP model—which shares a retinal disease phenotype with the model used for initial screening but involves different genetic and molecular mechanisms of degeneration.

**Reviewer #2 (Public review):**
Summary:In the manuscript entitled "Sex-specific attenuation of photoreceptor degeneration by reserpine in a rhodopsin P23H rat model of autosomal dominant retinitis pigmentosa" by Beom Song et al., the authors explore the transcriptomic differences between male and female wild-type (WT) and P23H retinas, highlighting significant gene expression variations and sex-specific trends. The study emphasizes the importance of considering biological sex in understanding inherited retinal degeneration and the impact of drug treatments on mutant retinas.Strengths:(1) Relevance to Clinical Challenges: The study addresses a critical limitation in inherited retinal degeneration (IRD) therapies by exploring a gene-agnostic approach. It emphasizes sex-specific responses, which aligns with recent NIH mandates on sex as a biological variable.(2) Multi-dimensional Methodology: Combining electroretinography (ERG), optical coherence tomography (OCT), histology, and transcriptomics strengthens the study's findings.(3) Novel Insights: The transcriptomic analysis uncovers sex-specific pathways impacted by reserpine, laying the foundation for personalized approaches to retinal disease therapy.

We are grateful for highlighting the strengths of our work.

Weaknesses:Dose OptimizationThe study uses a fixed dose (40 µM), but no dose-response analysis is provided. Sex-specific differences in efficacy might be influenced by suboptimal dosing, particularly considering potential differences in metabolism or drug distribution.

We acknowledge the limitation of using a fixed dose (40 µM) of reserpine in this study without conducting a comprehensive dose-response analysis. In the primary screens, the EC_50_ of reserpine was approximately 20 µM. We doubled the concentration for injection to account for the potential loss of reserpine during the in vivo procedures. As we observed the rescue effect of reserpine in mice, we used the same concentration for rats. The fixed-dose approach was chosen to maintain consistency with previous studies evaluating reserpine in retinal degeneration models and to facilitate comparison across studies. Efforts to identify optimal dosing were deprioritized, as the primary goal was different and this information cannot be directly translated to clinical applications.

We also agree that sex-specific differences in efficacy might be influenced by suboptimal dosing, particularly given potential variations in metabolism, drug distribution, and pharmacokinetics between male and female rats. However, recent pharmacokinetic studies on systemically administered reserpine in rats reported no statistically significant covariates, including body weight, age, breed, or sex, affecting pharmacokinetic (PK) or pharmacodynamic (PD) parameters (Alfosea-Cuadrado, G. M., Zarzoso-Foj, J., Adell, A., Valverde-Navarro, A. A., González-Soler, E. M., Mangas-Sanjuán, V., & Blasco-Serra, A. (2024). Population Pharmacokinetic–Pharmacodynamic Analysis of a Reserpine-Induced Myalgia Model in Rats. Pharmaceutics, 16(8), 1101. https://doi.org/10.3390/pharmaceutics16081101). Furthermore, no evidence of sex-specific differences in reserpine pharmacokinetics has been previously identified in available databases (National Center for Biotechnology Information (2025). PubChem Compound Summary for CID 5770, Reserpine. Retrieved January 13, 2025 from https://pubchem.ncbi.nlm.nih.gov/compound/Reserpine). Importantly, the drug in this study was administered intravitreally, where the ocular compartments are relatively isolated from systemic metabolism or excretion. Under these conditions, where absorption, distribution, metabolism, and excretion have minimal impact, we observed sex differences in efficacy using the same dose of drug.

Nonetheless, we agree with the reviewer and plan to pursue dose-response and other studies in future investigations.

Statistical AnalysisIn my opinion, there is room for improvement. How were the animals injected? Was the contralateral eye used as control? (no information in the manuscript about it!, line 390 just mentions the volume and concentration of injections). If so, why not use parametric paired analysis? Why use a non-parametric test, as it is the Mann-Whitney U? The Mann-Whitney U test is usually employed for discontinuous count data; is that the case here?Therefore, please specify whether contralateral eyes or independent groups served as controls. If contralateral controls were used, paired parametric tests (e.g., paired t-tests) would be statistically appropriate. Alternatively, if independent cohorts were used, non-parametric Mann-Whitney U tests may suffice but require clear justification.

We apologize for the lack of clarity. In line 124, we described the injection as “bilateral intravitreal injections of 5 µL of either vehicle or 40 µM reserpine,” and in Figure 1A, we annotated the bilateral injection as DMSO for both eyes and RSP for both eyes. To address this uncertainty, we added the clarification, “with each group receiving bilateral injections of either vehicle or reserpine” (lines 404–405). Since the results are not paired and involve continuous data for which the normality assumption cannot be confidently met or verified, we used the Mann-Whitney U test for statistical analysis.

Sex-Specific PathwaysThe authors do identify pathways enriched in female vs. male retinas but fail to explicitly connect these to the changes in phenotype analysed by ERG and OCT. The lack of mechanistic validation weakens the argument.The study does not explore why female rats respond better to reserpine. Potential factors such as hormonal differences, retinal size, or differential drug uptake are not discussed.It remains open, whether observed transcriptomic trends (e.g., proteostasis network genes) correlate with sex-specific functional outcomes.

We acknowledge that, while we identified pathways enriched in female versus male retinas, we did not explicitly connect these findings to the functional phenotypes measured by ERG and OCT. Although our transcriptomic data suggest that reserpine differentially influences pathways such as proteostasis and p53 signaling, we did not conduct mechanistic experiments to validate a causal relationship between these pathways and the observed outcomes.

In practice, designing a study to validate the mechanisms of a small molecule modulating multiple pathways presents significant challenges. If the pathways cannot be specifically modulated or if modulation could result in irreversible outcomes, the mechanistic validation becomes difficult to achieve. Drugs demonstrating mutation-agnostic efficacy are often investigated primarily through outcome measures and the analysis of affected pathways rather than through direct mechanistic validation (Leinonen, H., Zhang, J., Occelli, L. M., Seemab, U., Choi, E. H., L P Marinho, L. F., Querubin, J., Kolesnikov, A. V., Galinska, A., Kordecka, K., Hoang, T., Lewandowski, D., Lee, T. T., Einstein, E. E., Einstein, D. E., Dong, Z., Kiser, P. D., Blackshaw, S., Kefalov, V. J., Tabaka, M., … Palczewski, K. (2024). A combination treatment based on drug repurposing demonstrates mutation-agnostic efficacy in pre-clinical retinopathy models. Nature communications, 15(1), 5943. https://doi.org/10.1038/s41467-024-50033-5).

As recommended, we added potential factors that might influence the differential response to reserpine, based on other studies (lines 353–362) highlighting differences in dopamine storage capacity and estrogen independence. We also added a discussion on the possibility of sex-related differences in basal ERG response levels (lines 363–366).

**Recommendations for the authors:**

**Reviewer #1 (Recommendations for the authors):**
The study presents compelling findings on the neuroprotective effects of reserpine in a well-established model of retinitis pigmentosa (P23H-1). The use of ERG, optomotor assays, OCT, immunohistochemistry, and transcriptomic techniques provides a good exploration of the treatment's effects, particularly highlighting the differential response in females. The study underscores the potential of drug repurposing to expedite the availability of therapeutic interventions for patients.

Thanks for your generous comments.

While the manuscript presents an important contribution, I would like to highlight a few points that need clarification or further elaboration to strengthen the work:(1) Please include the photopic a-wave data in your analysis or provide a justification for its omission. Specifically, it would be valuable to know whether there is an improvement in this parameter under reserpine treatment.

We appreciate the reviewer’s suggestion to include photopic a-wave data in our analysis and acknowledge the importance of this parameter in evaluating cone photoreceptor function. However, we did not analyze the photopic a-wave amplitude in our study because we found the photopic a-wave has low amplitude and high variability, consistent with findings in other studies with P23H-1 rats (Orhan E, Dalkara D, Neuillé M, Lechauve C, Michiels C, et al. (2015) Genotypic and Phenotypic Characterization of P23H Line 1 Rat Model. PLOS ONE 10(5): e0127319. https://doi.org/10.1371/journal.pone.0127319) or even with wild type rats (V.L. Fonteille, J. Racine, S. Joly, A.L. Dorfman, S. Rosolen, P. Lachapelle; Do Rats Generate a Photopic a–Wave? . Invest. Ophthalmol. Vis. Sci. 2005;46(13):2246). We added the description (lines 435-437) explaining why the photopic a-wave was not analyzed. Studies with P23H-1 did not analyze the photopic a-wave, probably for similar reasons.

(2) In Figure 1, it would be helpful to include data from normal control animals to provide a benchmark for retinal degeneration in P23H-1 animals and to better contextualize the effects of reserpine treatment.

Thanks. As suggested, we have included data from normal control animals to Figure 1.

(3) The manuscript states that "Treated female retinas have significantly higher expression of the gene for P62 (SQSTM1), indicating a potential key route for reserpine's activity" (Line 331). Please explain how this difference in expression might translate into a better photoreceptor response in females compared to males.

The difference in P62 (SQSTM1) expression between treated female and male retinas could have important implications for the photoreceptor response. We have identified in our previous study that reserpine increased P62 that mediates proteome balance between ubiquitin-proteasome system (UPS) and autophagy. Together with the role of P62 in the regulation of oxidative stress, P62 might be important for photoreceptor survival and function. Higher expression of P62 in treated females could suggest more efficient cellular maintenance and a better ability to cope with stress, leading to improved photoreceptor survival and function.

(4) Numerous studies have shown that animal models of Parkinson's disease (e.g., those treated with MPTP or rotenone) or retinal tissue from Parkinson's patients exhibit dopaminergic cell death and associated vision loss. Please discuss how these findings relate to your results. Can you hypothesize how dopamine depletion by reserpine may lead to improved photoreceptor responses in your model?

We appreciate the reviewer’s insightful comments. Both MPTP and rotenone act via inhibition of complex I of the respiratory chain, causing cell death and leading to dopamine depletion. In contrast, reserpine acts by inhibiting the vesicular monoamine transporter, depleting catecholamines by preventing their storage and facilitating their metabolism by monoamine oxidase. Although reserpine and other agents can induce animal models of Parkinson's disease, reserpine differs from the others in several aspects: (i) reserpine do not induce neurodegeneration and protein aggregation; (ii) motor performance, monoamine content, and TH staining are partially restored after treatment interruption; and (iii) reserpine lacks specificity regarding dopaminergic neurotransmission (Leão, A. H., Sarmento-Silva, A. J., Santos, J. R., Ribeiro, A. M., & Silva, R. H. (2015). Molecular, Neurochemical, and Behavioral Hallmarks of Reserpine as a Model for Parkinson's Disease: New Perspectives to a Long-Standing Model. Brain pathology (Zurich, Switzerland), 25(4), 377–390. https://doi.org/10.1111/bpa.12253). We have discussed the various effects of catecholamine depletion on retinal diseases (lines 331–337). Both dopamine receptor antagonists and agonists, as well as catecholamine depletion, can exert protective effects on the retina. The reduction in scotopic b-wave amplitude observed at P54, followed by a lack of further progression in degeneration, may support the hypothesis that reduced neuronal activity due to catecholamine depletion could have mitigated damage to retinal neurons.

(5) For readers who may not be familiar with the P23H-1 mutation, it would be beneficial to include a brief description of the timeline and progression of retinal degeneration in this model.

As the progression varies among studies, we have provided our description on observations from the same facility where the animals were housed. The timeline and progression of retinal degeneration are briefly described in the results section (lines 112–115) and Supplementary Figure 1.

(6) Do you have any data on the effects of reserpine treatment in older animals? If available, this could provide additional insight into the potential applicability of reserpine in later stages of disease progression.

Unfortunately, we do not have data from older animals. As described in the results section (lines 116–124), we set the timepoint for interventions before functional impairment peaked, aiming to harness the remaining potential for rescue and promote functional improvement. Our approach focused on developing a gene-agnostic therapy that can delay disease progression and be delivered at an earlier stage than AAV-based therapies, using FDA-approved drugs.

(7) Molecular Basis of Sex Differences: The molecular mechanisms underlying the differential responses in males and females should be elaborated upon. If possible, include a discussion or hypothesis that addresses these sex-specific differences at the molecular level.

We thank the reviewer for highlighting the importance of addressing the molecular basis of sex-specific differences. In our study, we observed distinct transcriptomic responses to reserpine between male and female rats, particularly in molecular pathways related to proteostasis and p53 signaling. While the sex-specific differences in these molecular pathways remain to be fully evaluated, we have added a discussion on sex differences in reserpine responses, incorporating findings from other studies (lines 353–366).

**Reviewer #2 (Recommendations for the authors):**
(1) There is no mention in the manuscript about the fact that the transgene rats have several copies of rhodopsin and how this can affect these sex differences. Would it be the same in the P23H KO mouse? Or in other models with a single copy of the mutation?

We have described in the Materials and Methods section how they were bred, but we did not specifically mention the allele status in the manuscript. Hemizygous P23H-1 rats used in this study carry a single P23H transgene allele with a transgene copy number of 9, in addition to the normal two wild-type opsin alleles. We added this description to clear the uncertainty (lines 384-387).

(2) This sentence: in abstract lines 26 to 29: "Recently, we identified reserpine as a lead molecule for maintaining rod survival in mouse and human retinal organoids as well as in the rd16 mouse, which phenocopy Leber congenital amaurosis caused by mutations in the cilia-centrosomal gene CEP290 (Chen et al. eLife 2023;12:e83205. DOI: https://doi.org/10.7554/eLife.83205)", to my vew, does not belong to the abstract, maybe in the introduction as stage of art.

Thank you for asking. According to the guidelines for the research advance articles (that follow previously published studies), a reference to the original eLife article should be included in the abstract. As specified in the guidelines, we have updated the citation format to (author, year) for referencing eLife articles (line 29).

(3) Lines 167-170: "Histologic evaluation of the retinas also demonstrated more prominent ONL thinning in the dorsal retina and increased ONL thickness in the dorsal retina measured at 1,000, 1,250, and 1,500 µm distant from the optic nerve head in reserpine-treated group compared with control group (Figure 3C)". I do not understand this sentence. Is it a more prominent thinning or an increased thickness?

We apologize for the confusion caused by this sentence. The histological evaluation showed that ONL thinning was more pronounced in the dorsal retina of control group, which was consistent with OCT findings in Figure 3A. Reserpine treatment increased the ONL thickness in the dorsal retina at specific distances from the optic nerve head (1,000, 1,250, and 1,500 µm). We have revised the sentence for clarity (lines 165-168).

(4) Lines 182-185 and Figure 4B: FL is not the best approach to quantify rhodopsin levels. Since the DAPI staining is overexposed, it is hard to evaluate the staining of RHO in the ONL. From the visible staining in the OS, it is only possible to affirm that the OS are longer in RSP-treated retinas... more is not to be affirmed based on these figures. I suggest using WB.

We acknowledge the reviewer’s concern regarding the use of fluorescence imaging to quantify rhodopsin levels. While our current data highlight structural preservation, such as the length of the outer segments, we agree that drawing conclusions about rhodopsin levels from fluorescence staining is limited. As we do not have samples for WB and fluorescence imaging cannot quantify rhodopsin, we have revised the description (lines 180-184).

(5) Lines 188-190 and Figure 4C: The images in 4C showed an extreme divergence between treated and untreated retina concerning the amount of stained cones, which is not observed at the quantification at 1000µm statistic. Are the images not representative?

We agree with the reviewer that the images in Figure 4C may not adequately represent the quantified data. To address this, we have changed the figure to reflect the quantification results accurately.

(6) Figures 6C-6D and 6G. Why do the authors not use any statistical analysis? Or are the differences not statistically significant? Why do authors use only WT and DMSO controls? What about untreated P23H controls (no DMSO)?

Thanks for checking, and we apologize for the oversight. We have updated figures 5, 6 and S5 to include adjusted p-value in relevant plots. In addition, details of significance threshold are available in supplementary tables. Regarding controls, untreated P23H retinas (without DMSO) were not included in the current analysis, as our experience shows that DMSO injection itself does not cause functional or structural changes. The key data demonstrating the effect of reserpine involve a comparison between the group treated with reserpine and the control group treated with DMSO, as the only difference between these groups is the involvement of the drug.

(7) Validation of findings by testing key genes (e.g., p62/SQSTM1, Nrf2) using qPCR or immunohistochemistry will strengthen the findings.

We appreciate the reviewer’s suggestion to validate key findings using qPCR or immunohistochemistry, as such experiments are crucial for further strengthening our conclusions. While this was not feasible in the current study due to various constraints, we fully recognize their importance and plan to incorporate these in our follow-up studies.